# Alternative Splicing as a Modulator of the Interferon-Gamma Pathway

**DOI:** 10.3390/cancers17040594

**Published:** 2025-02-10

**Authors:** Parul Suri, Ariana Badalov, Matteo Ruggiu

**Affiliations:** 1College of Pharmacy and Health Sciences, St. John’s University, 8000 Utopia Parkway Queens, New York, NY 11439, USA; parul.suri13@my.stjohns.edu; 2Laboratory of RNA Biology and Molecular Neuroscience, Department of Biological Sciences, St. John’s University, 8000 Utopia Parkway Queens, New York, NY 11439, USA; ariana.badalov22@my.stjohns.edu

**Keywords:** IFN-γ, alternative splicing, cancer, immunity, therapy

## Abstract

This review examines how a crucial immune system protein called interferon-gamma (IFN-γ) is affected by alternative splicing—a process where genes can produce multiple versions of proteins through different arrangements of genetic material. While IFN-γ is known to be important for fighting cancer and infections by activating immune cells, how alternative splicing impacts its function is not fully understood. The review discusses how this process can either enhance or suppress IFN-γ’s effects on the immune system by creating different variants of proteins involved in its signaling pathways, both upstream and downstream. Understanding how alternative splicing may selectively modulate IFN-γ’s function could lead to the development of new treatments for cancer, autoimmune diseases, and infections.

## 1. Introduction

Interferon-gamma (IFN-γ), the sole member of the type II interferon family, is a crucial cytokine in immune system regulation, particularly in cancer immunosurveillance. This review explores its functionality through the lens of alternative splicing—a molecular process generating multiple RNA and protein variants from a single gene.

By examining upstream regulators and downstream targets of IFN-γ, this review reveals how subtle changes in RNA splicing can dramatically alter immune cell function, expanding our understanding of immunotherapy strategies and offering new insights into disease progression and therapeutic strategies.

## 2. Interferon-Gamma: Structure and Function

IFN-γ plays a pivotal role in immune system regulation and cellular defense mechanisms. It is a critical immune mediator with potent antitumor properties. Unlike other interferons, IFN-γ has a unique structure and plays a distinct function in cancer immunosurveillance by directly inhibiting tumor cell proliferation, enhancing immune cell activation, and promoting potent anti-tumor immune responses. Its ability to stimulate macrophages, activate natural killer cells, and induce major histocompatibility complex (MHC) expression makes it a key cytokine in the body’s defense against cancer progression, highlighting its significance beyond its initial characterization as an antiviral agent [1,2].

Structurally, IFN-γ is a homodimeric protein composed of two 17 kDa polypeptides that undergo extensive N-glycosylation, ultimately assembling non-covalently in a distinctive antiparallel fashion to form a 50 kDa molecule [3]. This unique symmetry and molecular architecture allow for an intriguing receptor-binding mechanism, where two receptors can be bound by a single IFN-γ molecule simultaneously, with potential cross-talk involving IFN-α/β receptors to enhance IFN-γ signaling and its biological effects [1].

### 2.1. Cellular Production and Genetic Regulation

The production of IFN-γ is predominantly associated with activated lymphocytes, including CD4 T helper type 1 (Th1) cells, CD8 cytotoxic T cells, γδT cells, and natural killer (NK) cells, which secrete IFN-γ in greater amounts than B cells, other antigen-presenting cells (APCs), and NK T cells. IFN-γ exerts a profound influence on gene expression, regulating the expression of hundreds of genes involved in critical cellular processes such as transcriptional activation, apoptosis, cell cycle regulation, and inflammatory signaling. These genes are expressed differently in different cells, mediating the pleiotropic effects of IFN-γ [4]. T-lymphocytes are the primary paracrine source of IFN-γ in adaptive immunity, whereas autocrine IFN-γ produced by APCs helps sustain self and neighbor cell activation, a crucial mechanism for early control of pathogen spreading [5,6]. IFN-γ released by APCs and NK cells is linked to the modulation of autocrine function as well as the initial defense against the host [1]. Notably, constitutive expression of type I and II IFN is strictly regulated during physiological conditions, staying localized in tissues and having no systemic effects. For example, constitutive expression of endogenous IFN-γ plays crucial roles in bone formation, immune cell function homeostasis, and maintenance of the hematopoietic stem cell niche [7,8,9].

### 2.2. Immune Response Modulation

IFN-γ displays remarkable versatility in enhancing the host’s immune response against microbes and potential threats [10,11]. Its immune-boosting capabilities include a complex array of functions that extend beyond simple pathogen resistance: it enhances the production of reactive oxygen species (ROS) and reactive nitrogen intermediates (RNIs), elicits antiviral reactions, and amplifies antigen production via APCs, alongside enhancing antigen detection through complementary T-cell contact [2]. Interestingly, this process has additionally been shown to be essential for cancer immune monitoring, boosting immunity against cancer and aiding in the identification and removal of tumors [1].

### 2.3. Antitumor Immunity

In the field of oncology, IFN-γ has emerged as a fundamental mediator of antitumor immunity [12]. Its multifaceted antitumor activities include modulating both neoplastic cells and immune effector cells. For instance, IFN-γ enhances the expression of MHC class I molecules on the surface of tumor cells, thereby augmenting their antigenicity through increased antigen presentation [13,14]. Furthermore, IFN-γ amplifies the cytotoxic functions of NK cells and cytotoxic T lymphocytes (CTLs) [15]. These mechanisms act synergistically to render tumor cells more vulnerable to detection and elimination by IFN-γ-activated immune effector cells, a property ascribed to enhanced presentation of tumor-associated antigens [12,16]. IFN-γ exhibits direct antitumor effects on tumor cells, including the inhibition of cellular proliferation via cell cycle arrest, mediated through the activation of the tumor suppressors p21 and p27 [17,18]. Additionally, IFN-γ is recognized for its antitumor role through the induction of apoptotic cell death across various cancer cell types [19]. Moreover, IFN-γ has the capacity to initiate necroptosis, a necrosis-like regulated cell death that occurs through the activation of RIP1 serine-threonine kinase [20]. The inhibition of tumor angiogenesis represents another established antitumor activity of IFN-γ. The secretion of IFN-γ by mesenchymal stromal cells within the tumor microenvironment has been demonstrated to repolarize tumor-associated macrophages towards the M1 inflammatory phenotype, resulting in the reduction of tumor burden in a neuroblastoma model [21]. Interestingly, while traditionally associated with facilitating the development of immunosuppressive regulatory T cells (Treg), recent findings indicate that IFN-γ also enhances antitumor immunity and tumor eradication by promoting Treg fragility [22].

Even though IFN-γ’s status as a conventional antitumor immune factor is widely recognized and has been meticulously characterized, its historical potential as a protumor factor is also well-documented [23]. A recurring theme that appears to emerge from studies on the protumorigenic effects of IFN-γ is that tumors subjected to IFN-γ exhibit enhanced immunoevasive advantages [13]. Prolonged exposure to elevated levels of IFN-γ may therefore impose selective immune pressure on tumor cells, potentially leading to a reduction or complete loss of MHC class I and other critical antigen presentation genes. This mechanism has been observed in specific cancer cell lines like M14 melanoma and CT26 colon cancer cell lines, wherein exposure to IFN-γ led to a downregulation of antigen presentation and facilitated immune evasion [24,25].

### 2.4. Signaling Mechanisms

IFN-γ signaling mechanisms represent a complex and tightly regulated communication system. It is primarily mediated through its heterodimeric receptor (IFN-γR) complex. IFN-γ binds to IFN-γR on target cells including dendritic cells (DCs) and macrophages, among others, to produce its physiological effects. When IFN-γ binds to its receptor—comprising the ligand-binding alpha subunit (IFNGR1) and the signal-transducing beta subunit (IFNGR2)—it triggers a complex signaling cascade of cellular responses [2,26,27,28]. The binding of IFN-γ to its receptor triggers a conformational change that brings the two receptor subunits, IFNGR1 and IFNGR2, closer together and activates the associated Janus kinases (JAK1 and JAK2) through autophosphorylation [29,30,31]. Subsequently, JAK1 and JAK2 phosphorylate the signal transducer and activator of transcription 1 (STAT1) [29]. Phosphorylated STAT1 homodimerizes, translocates to the nucleus, and facilitates the transcription of IFN-γ-related genes via gamma-activated sequences (GAS) promoter motifs [2,32]. This process is critical for cell differentiation. For instance, during T cell differentiation into the Th1 lineage, STAT1 phosphorylation and IFN-γ-IFN-γR signaling directly triggers the key initial step in the activation of the transcription factor T-bet [32]. This regulatory complexity is further enhanced by the recruitment of a second factor, Runx3, which binds to the IFN-γ promoter and simultaneously silences the IL-4 gene expression [33]. As a result, IFN-γ is essential for preventing Th1 cells from generating too much IL-4 and for preserving T-bet expression over time [1,2,34].

This review analyzes research describing the alternative splicing of a subset of IFN-γ-controlling and IFN-γ-regulated factors, as well as their role as novel biomarkers and therapeutic entry points. It required approximately thirty years to consider IFN-γ as a possible candidate for immunotherapy against tumors after Wheelock discovered that this cytokine suppressed the replication of viruses in 1965 [35,36]. Consequently, this review aims to provide a more thorough understanding of IFN-γ’s functions through the lens of alternative splicing, which has the potential to inform the planning and administration of novel clinical immunotherapy strategies.

## 3. Alternative Splicing

The sequencing of the human genome revealed that we have ~22,000 protein-coding genes, an unexpectedly low number that is on the same scale as the number of genes in less complex organisms such as worms (~19,000 genes) and flies (~14,000 genes), but also significantly lower than the number of genes present in many plants (~40,000 genes in rice) [37,38,39,40]. This lack of a correlation between the number of genes in an organism and phenotypic complexity suggests that the number of genes in a certain organism is not sufficient, by itself, to explain how such complexity is generated. Alternative splicing, a post-transcriptional mode of gene regulation by which multiple mature RNAs (mRNAs) are generated from a single gene [41], has been suggested as a possible solution to this paradox, as more complex organisms such as mammals display a higher frequency of alternative splicing compared to less complex organisms such as worms and flies [42,43,44,45,46,47]. Transcriptome analyses have demonstrated that alternative splicing is present in most eukaryotes and that the percentage of genes and exons undergoing alternative splicing is higher in vertebrates compared to invertebrates [48]. For instance, alternative splicing occurs in 90–95% of human genes in a tempo-spatial-dependent pattern that relies on *trans*-acting RNA-binding proteins (or splicing factors) binding to *cis*-acting elements on the precursor messenger RNA (pre-mRNA) [44,49,50]. Therefore, by allowing for the synthesis of many mature mRNAs with various functions from just one gene, alternative splicing provides a significant boost in proteome complexity and diversity without the need for a proportional increase in genome size [51]. While, in bacteria, every cistron may generate a polypeptide that differs entirely in sequence and function from the other cistrons of the same gene, in eukaryotes, the different polypeptides generated from one gene via alternative splicing are usually similar but not identical, containing conserved and distinct regions that may result in mild or substantial changes both at the mRNA and protein levels [47].

The view that exons encode proteins and introns do not is an oversimplification [47]. For example, splicing happens in non-coding RNA as well as genes that encode mRNAs [52,53]. Furthermore, in the context of mRNAs, the initial AUG (the start codon of a translated open reading frame) may be incorporated further downstream into the first or even the second exon; this also holds true for non-AUG start codons, which have been found to occur more frequently than predicted [54].

Alternative transcription start sites and alternative polyadenylation sites on the first and last exons, respectively, as well as five other major alternative splicing modes (exon skipping or cassette exons, intron retention, mutually exclusive exons, alternative 5′ splice site, and alternative 3′ splice site) can result in different transcripts. More complex, combinatorial alternative splicing events, such as cassette exons with alternate 5′ or 3′ splice sites, can result from multiple combinations of these binary events, either at the same site or at different sites (Figure 1).

Numerous gene transcripts, in fact, undergo multiple alternative splicing events at different sites along the pre-mRNA that, if not precisely synchronized, may result in combinatorial diversity [51,55]. There are significant differences in the overall amount of alternative splicing and the relative relevance of each type of alternative splicing event throughout eukaryotic lineages [55]. For instance, the most prevalent type of alternative splicing in metazoans is exon skipping, but intron retention is more common in fungi and plants [56,57]. In addition to changing transcript abundance, alternative splicing can also change the structure, stability, and turnover of transcripts and the proteins they encode. This allows for the functional diversification of many different protein isoforms derived from a single gene, as well as their intracellular localization, binding characteristics, enzymatic activity, and protein stability [47,51,58,59].

Nearly all polymerase II transcripts undergo splicing, and pre-mRNA splicing is disrupted in up to 50% of disease-causing genetic mutations [60,61,62,63,64]. One of the main sources of protein complexity in the neurological and immunological systems is alternative splicing [64,65,66]. These tissues are made up of a very varied range of cell types that need to be able to process large amounts of information, adapt, and control cellular activity and differentiation quickly and precisely. Through alternative splicing and nonsense-mediated decay (NMD), one can modify protein function, alter cytokine signaling (e.g., certain isoforms of cytokine receptors frequently have antagonistic functions in the signaling of a particular cytokine), and down-regulate gene expression by producing unstable or non-functional protein and mRNA isoforms [47]. Furthermore, the accumulation of somatic mutations in cancers leads to the creation of tumor-specific antigens (TSAs), also known as tumor neopeptides or neoantigens, which are unique protein fragments produced by the mutated cancer cells that can be recognized by the immune system as foreign and targeted for destruction [67]. This, in turn, can elicit an anti-cancer immune response in patients [68,69], and neoantigens are promising therapeutic targets for cancer treatment [70]. However, even though neoantigens can also be generated through aberrant RNA splicing [68,71,72], and splicing-modulating therapeutic strategies have been investigated for the last two decades [73,74,75,76,77], the contribution of aberrant splicing to the generation of neoantigens in cancer has been largely overlooked, and its therapeutic potential not fully exploited [78,79,80,81,82,83,84].

The dysregulation of alternative splicing is a key molecular hallmark of cancer [85], and somatic mutations leading to aberrant alternative splicing may impact the development of cancers [86,87] and might interfere with the pathways that proteins interact with to form tumors [59]. However, how alternative splicing may modulate the IFN-γ pathway and its function is still poorly understood. In mammals, the *IFNG* gene has only four exons and does not appear to be alternatively spliced. In this review, therefore, we focused on various regulators that are located upstream of IFN-γ and undergo alternative splicing to modulate the activity of IFN-γ (Table 1 and Figure 2), as well as the downstream effectors of IFN-γ activity that are also alternatively spliced (Table 2 and Figure 1). Recent advances in therapeutics targeting splicing catalysis and splicing factors are discussed as well.

## 4. Upstream Regulators of IFN-γ and Their Associated Co-Factors

### 4.1. Tumor Neopeptides

Tumor neopeptides, or neoantigens, are antigens generated de novo by tumor cells as a result of various tumor-specific alterations including dysregulated RNA splicing. As such, tumor neopeptides are very promising targets for cancer immunotherapy [70]. Cells in uveal melanoma undergo aberrant alternative splicing to produce tumor neopeptides such as AMZ2P1-neo and MZT2B-neo. The most frequent alternative splicing event observed is exon skipping, although alternative 5′ splice sites, alternative 3′ splice sites, alternative first exons, and retained introns have also been reported. These splice variants, validated through long-read sequencing and mass spectrometry, were shown to stimulate IFN-γ production in CD8^+^ T cells, which enhanced the killing of uveal melanoma cells. The mechanism by which these tumor neopeptides increase IFN-γ expression is not known. While SF3B1, a splicing factor frequently mutated in cancers [88,89,90,91,92,93,94,95,96,97], is often targeted in therapies, these neoepitopes were present regardless of SF3B1 mutation status, suggesting an SF3B1-independent mechanism of action. This work expands the potential therapeutic benefit to a broader group of uveal melanoma patients, making these shared neoepitopes promising targets for immunotherapy [98].

### 4.2. IRF8

Interferon regulatory factor 8 (IRF8) is a transcription factor critical for normal hematopoiesis and myeloid cell development [99,100,101]. The human *IRF8* gene spans 23 kb of chromosome 16q24.1, has nine exons and eight introns, and it encodes a 426 amino acids protein. Constitutively expressed by lymphoid and myeloid cell lines, IRF8 is additionally activated by IFN-γ and found in intestinal, skin, lung, liver, ocular lens, cornea, and heart epithelial cells [102,103,104], and is required for the growth, development, and maturation of DC1 (cDC1) and pDC lineages [105,106,107]. There are three IRF8 splice variants (IRF8-SVs). Every splice variant splices out exon 1 of the standard sequence while introducing a fragment of the terminal region of intron 1 into the transcript. Additionally, a subgroup of acute myeloid leukemia (AML) patients express IRF8-SVs at substantially greater levels (>two-fold) than hematopoietic cells from healthy donors, despite the fact that these splice variants are present in the hematopoietic cells of a normal adult at extremely reduced levels [108,109], and loss of IRF8 inhibits the growth of AML cells, suggesting a dual role for IRF8 as a prognostic factor and therapeutic target in AML [110].

### 4.3. TAp63

The transcription factor p63 (Tap63), a homolog of the p53 tumor suppressor, and the Δ133p53 isoform play significant roles in regulating IFN-γ signaling in different breast cancer subtypes. TAp63 is shown to drive the expression of IFN-γ-related genes in ER- (estrogen receptor-negative) wild-type TP53 (wtTP53) breast cancers, where it binds to promoters of IFN-γ signaling genes. In contrast, in ER+ mutant TP53 (mTP53) tumors, the Δ133p53 isoform takes over this regulatory role. The Δ133p53 isoform competes with TAp63 and TAp73, inhibiting their function in these breast cancer subtypes. Therefore, Δ133p53 indirectly downregulates IFN-γ signaling by impairing the transcriptional activation usually mediated by TAp63 in certain breast cancer subtypes [111]. The distinct expression of IFN-γ pathway genes mediated by p63 or Δ133p53 is subtype-dependent, and reduced IFN-γ signaling is associated with poor prognosis across different breast cancer subtypes. These findings highlight the complex interplay between TP53 family isoforms in modulating immune responses via the IFN-γ pathway, suggesting potential therapeutic strategies for targeting these mechanisms in breast cancer treatment [11,112].

### 4.4. FOXP3

Forkhead Box P3 (FOXP3) is a lineage-defining master transcription factor that regulates the expression of CD4^+^CD25^+^ thymus-derived Tregs [113], a subset of T cells that mediate the immune response against antigens and inhibit conventional T-cell activation and proliferation [114,115]. In human T cells, FOXP3 is expressed as two predominant isoforms: the full-length isoform, FOXP3FL, and an alternatively spliced isoform, FOXP3Δ2, that skips exon 2 which encodes a region mapping within the proline-rich part of the protein [116,117]. FOXP3Δ2 has enhanced nuclear retention due to the absence of a nuclear export sequence (NES) located in exon 2. This affects its ability to interact with certain cofactors [115,118]. Exon 2 skipping causes a conformational change in FOXP3 that results in relieving autoinhibition of the forkhead domain (FKH) and enhancing its DNA-binding capacity. This increased binding can potentially downregulate cytokine expression such as IFN-γ [118,119]. Foxp3Δ2 differentially regulates the homeostasis of tTregs and peripherally-induced Tregs (pTregs). This alternative splicing event alters the functionality of Foxp3, influencing its ability to control immune responses [119,120,121]. For example, FOXP3FL can interact with transcription factors of the ROR family via its exon 2-encoded LXXLL motif, and this interaction inhibits ROR-mediated transcription of inflammatory cytokines. FOXP3∆2 lacks this interaction activity, resulting in altered cytokine expression profiles [118,119,121,122,123]. Foxp3Δ2-bearing tTregs are shown to be less responsive to TCR and IL-2 stimulation, which can reduce their suppressive capacity. In contrast, Foxp3Δ2 pTregs exhibit a competitive advantage in environments like the gut, where they adapt to local cues [119]. Given that Tregs are critical in limiting excessive IFN-γ production by effector T cells, the differential regulation of tTregs and pTregs by Foxp3Δ2 could significantly impact the balance of pro-inflammatory and regulatory signals, including IFN-γ, in various tissues [124]. The dysregulation of this balance could contribute to either heightened immune activation or immune tolerance, depending on the context, making the Foxp3Δ2 isoform an important factor in immune homeostasis and anti-tumor immunity [119,120]. Co-expression of both FOXP3FL and FOXP3Δ2 is crucial for optimal Treg cell function [121]. The combined expression of these isoforms enhances the suppressive capabilities of Treg cells and modulates cytokine production, including IFN-γ. Specifically, the co-expression leads to a significant reduction in IFN-γ production, suggesting a synergistic role in maintaining immune homeostasis [121]. Alongside FOXP3Δ2, two other isoforms, FOXP3Δ2Δ7 and FOXP3Δ7, where exon 7 maps to a region forming part of a leucine-zipper structure [116,125], have also been described in human lymphocytes and epithelial cells [121,126,127,128]; however, unlike FOXP3FL and FOXP3Δ2, neither appears to play a role in Treg suppressive activity [129]. These findings indicate that Tregs may be able to differentiate in the presence of FOXP3Δ2 and in the absence of FOXP3FL, but that expression of FOXP3Δ2 alone may lead to impaired suppressive function and systemic autoimmune disease which cannot be rescued from the absence of FOXP3FL. Collectively, the data suggest that both FOXP3FL and FOXP3Δ2 isoforms may be necessary for Tregs to fully acquire suppressive functionality [128]. The expression of both major FOXP3 variants (FOXP3FL and FOXP3∆2) appears to be essential for the normal function of Tregs [119,128,130], and failure or imbalance of FOXP3 splice variant expression leads to severe and multi-organ auto-immune syndromes [119,120,130,131,132,133]. Furthermore, a FOXP3 splice site mutation causing exon 7 skipping and two different mutations in FOXP3 exon 7 have been identified in patients with immune dysregulation, polyendocrinopathy, enteropathy, and X-linked (IPEX) syndrome, a rare genetic disorder that causes severe autoimmune disease in children [132,134,135], suggesting that exon 7 has an important role in the suppressive function of FOXP3 [121]. Strategies targeting FOXP3 splicing isoforms in Tregs may provide potential new therapeutic approaches for the treatment of autoimmune diseases, inflammation, and cancer [118].

### 4.5. SRSF1

Serine/arginine-rich splicing factor 1 (SRSF1) plays a critical role in regulating IFN-γ production by modulating the splicing of RhoH, a GTPase involved in T-cell receptor signaling [136]. In SRSF1-deficient T cells, the reduced expression of RhoH leads to heightened activation of T cells, which results in an overproduction of IFN-γ and IL-17, exacerbating inflammatory conditions such as nephritis [136]. SRSF1 acts as a splicing regulator, ensuring the proper expression of RhoH, which limits T cell activation and suppresses excessive immune responses [137]. This regulation helps mitigate inflammation and tissue damage, especially in autoimmune diseases where IFN-γ plays a pivotal role. When RhoH expression is reduced, the inhibitory effects on TCR-proximal signaling are lost, leading to heightened activation of the TCR pathway. This results in increased differentiation of Th1 cells and elevated production of IFN-γ [138]. This work highlights the importance of SRSF1 in maintaining immune homeostasis and suggests potential therapeutic avenues for controlling immune-mediated diseases by targeting splicing mechanisms [136,139].

### 4.6. RBM39

The serine/arginine-rich RNA-binding protein RBM39 plays a significant role in regulating IFN-γ through its influence on the splicing and transcription of key immune factors [140]. This regulation is particularly evident in its interaction with interferon regulatory factor 3 (IRF3), which is crucial for the innate immune response. A recent preprint shows how RBM39 modulates the splicing of IRF3, impacting the expression of interferon-stimulated genes (ISGs) in response to various stimuli, including viral infections and lipopolysaccharide (LPS) exposure [141]. A recent study provides insights into the role of RBM39 in splicing and its autoregulation mechanisms, which may indirectly relate to broader regulatory functions in gene expression [142]. The study highlights that all four domains of RBM39 contribute to its function in splicing. The serine/arginine-rich RS domain is essential for RBM39’s role in splicing, as mutants lacking this domain could not rescue splicing events. RBM39 appears to have a different RNA-binding selectivity compared to U2AF2, which is important for understanding how RBM39 modulates gene expression during pre-mRNA splicing. This work reveals how RBM39 autoregulates its expression by enhancing the inclusion of a poison exon in its own pre-mRNA via a negative feedback loop. This autoregulation is crucial for maintaining RBM39 levels and could have implications for its role in cancer biology [142]. The research also uncovers that RBM39 actively participates in splice site selection, which is vital for proper mRNA processing and could influence the expression of various genes, potentially including those involved in immune responses like IFN-γ. RBM39 mutations affect its ability to bind RNA and regulate splicing, resulting in downstream effects on the expression of genes that are critical for immune functions. Thus, this work provides significant insights into the role of RBM39 in splicing and autoregulation, which may have broader implications for gene expression regulation in various biological contexts. Because of its vital role in tumorigenesis and its broad development prospects in clinical treatment and drug research, RBM39 is a promising therapeutic target for cancer [143,144,145].

## 5. Downstream Effectors of IFN-γ Activity

### 5.1. IFNGR1

IFN-γ binding to its heterodimeric receptor results in the activation of JAK1 and JAK2 kinases, which are pre-associated with the cytoplasmic domain of the IFNGR subunits. STAT1 is then recruited to the complex as a homodimer and is phosphorylated, resulting in the formation of IFN-γ activation factor (GAF) [146,147,148]. GAF then translocates to the cell nucleus to bind GAS sequences on promoters and trigger IRF1 transcription [149,150]. IRF1 is necessary for the IFN-γ-mediated response of human macrophages towards mycobacteria [151]. Interestingly, a recent report describes a patient with a homozygous variant of *IFNGR1* (NM_000416.2, c.861 + 2T > A) that, by disrupting the invariant GT dinucleotide at the 5′ splice site in intron 6 is predicted to cause exon 6 skipping. This mutation appears to be loss-of-function, as *IFNGR1* expression is undetectable in this patient [152]. The pathogenic mechanism underlying this mutation was not investigated.

### 5.2. STAT1

The signal transducer and activator of transcription 1 (STAT1) are essential for IFN-mediated immunity and undergo alternative splicing to generate two isoforms, STAT1α and STAT1β, which exhibit distinct functional properties in immune responses [153,154,155,156]. STAT1 is alternatively spliced into full-length STAT1α, which has a C-terminal transactivation domain (TAD), and C-terminally truncated STAT1β, which lacks this domain and is often considered transcriptionally inactive [153,154,155,157,158,159]. The roles of the spliced isoforms of STAT1, specifically STAT1α and STAT1β, in regulating IFN-γ signaling and responses have been investigated in several studies [160,161,162]. Contrary to previous beliefs that STAT1β is transcriptionally inactive, the study reveals that STAT1β is capable of inducing a significant number of IFN-γ-responsive genes, although often at lower levels compared to STAT1α [156,162]. STAT1β, in fact, displays a weaker transcriptional activity compared to STAT1α in mediating IFN-γ signaling [156]. This indicates that STAT1β is not merely a dominant-negative regulator but has its own transcriptional activity in response to IFN-γ. The research shows that while both isoforms can regulate many of the same genes, they exhibit distinct transcriptional profiles [156,162]. Some genes strictly require STAT1α for their expression, while STAT1β can limit the expression of a small subset of genes, particularly at later time points after stimulation with IFN-γ [156,162]. In the absence of STAT1α, STAT1β demonstrates prolonged phosphorylation and binding to promoters of IFN-γ-responsive genes [156]. This prolonged activity suggests that STAT1β can maintain its functional role in the absence of STAT1α, contributing to the immune response. The transcriptional activity of STAT1β is linked to functional biological responses, such as IFN-γ-induced antiviral activity in vitro and antimicrobial immunity in vivo [156,163,164]. This highlights the importance of STAT1β in mediating immune responses, particularly against pathogens. The study notes that while STAT1β can induce genes in response to IFN-γ, the induction occurs with a delay compared to STAT1α, suggesting that while both isoforms are functional, they contribute differently to the timing and magnitude of the immune response [156].

As previously mentioned, STAT1α and STAT1β differ in their C-terminal TAD domain, which is present in STAT1α but absent in STAT1β. This difference is crucial as it influences their transcriptional activity in response to cytokines like IFN-γ [156,162]. STAT1β lacks the C-terminal TAD and has been reported to be a weaker transcriptional activator compared to STAT1α. However, this activity is highly dependent on the specific target gene being regulated. The study highlights that the C-terminal TAD of STAT1 is essential for the recruitment of RNA polymerase II (Pol II) and for establishing active histone marks at certain promoters, such as the class II major histocompatibility complex transactivator (CIIta) promoter. In contrast, the C-terminal TAD is not required for the induction of IRF7, which is also mediated by STAT1 in response to IFN-γ. The research indicates that STAT1β exhibits gene-specific transcriptional activity, which varies from being completely impaired to showing increased activity at different time points after IFN-γ treatment. This suggests that the isoform’s effectiveness can change based on the gene context. The study also reveals that the C-terminal TAD is necessary for the efficient recruitment of components of the Mediator complex to the promoters of IRF1 and IRF8, which are important for the transcriptional response to IFN-γ [162,165].

### 5.3. IRF1

The tumor microenvironment alters the immune response by modulating the splicing of interferon regulatory factor 1 (IRF1), a key transcription factor in Th1 cells, resulting in the production of different IRF1 protein isoforms with altered activity [166]. The TGF-β present in the tumor microenvironment induces the production of IRF1Δ7, an alternative splicing isoform that lacks exon 7, through the action of the SFPQ splicing factor. IRF1Δ7 acts as a dominant-negative isoform that competes with full-length IRF1, reducing its ability to activate downstream targets that are essential for IFN-γ production, such as IL-12rb1, leading to reduced IFN-γ secretion in Th1 cells. As IFN-γ is crucial for anti-tumor responses, this splicing event compromises the immune system’s ability to mount an effective response against tumors. Targeting IRF1Δ7 to restore full-length IRF1 expression could potentially enhance Th1-mediated anti-tumor immunity [166].

In breast epithelial cells, IRF1 plays a critical role in the regulation of alternative splicing of the carcinoembryonic antigen-related cell adhesion molecule 1 (CEACAM1), particularly inducing the long (L) isoform that contains an immunoreceptor tyrosine-based inhibition motif (ITIM). CEACAM1 pre-mRNA undergoes alternative splicing, producing either a short (S) isoform, which is more commonly expressed in epithelial cells, or the L-isoform, which is predominant in immune cells [167]. IRF1, through binding to an interferon-stimulated response element (ISRE) in the *CEACAM1* promoter, drives the splicing switch to produce the L-isoform. This splicing event is crucial as the L-isoform, due to its ITIM, contributes to immune regulation by inhibiting immune cell activation, which can impact tumor progression. It has been reported that inflammatory signals like IFN-γ, acting through IRF1, play a significant role in promoting this splicing switch, particularly at the invasive front of breast tumors, suggesting that the tumor microenvironment can modulate immune evasion through this mechanism [168]. Interestingly, IRF1 and a variant of heterogeneous nuclear ribonucleoprotein L (Lv1) coordinately regulate *CEACAM1* transcription, alternative splicing, and translation, and may significantly contribute to *CEACAM1* silencing in cancer [169].

IRF1 becomes ineffective in a variety of cancers through a number of processes, such as the removal of chromosome 5q31’s IRF1 gene-encoding region, binding of nucleophosmin, a candidate ribosome assembly factor, the deactivation of the tumor suppressor function by the human papillomavirus 16 oncoprotein E7, and the expression of IRF2. Exon skipping in IRF1 gene transcripts has been documented in a number of investigations including bone marrow and peripheral blood mononuclear cells from healthy donors, patients with myelodysplastic syndrome, and patients with chronic myeloid leukemia [170,171,172,173]. The majority of the mutant species lacked either exon 2 or both exons 2 and 3. Since these two splice variants lack the DNA binding domain, they cannot operate as transcription factors and do not seem to have an impact on how wild-type IRF1 activates genes. Five different IRF1 splice variants from human cervical cancer tissue samples that resulted from skipping exons 7, 8, or 9, or a combination of those exons have been detected and described that are both malignant and benign. In contrast to the previously discovered alternative splice variants, these variants do not exhibit transcriptional activity; instead, they seem to compete with wild-type IRF1, thus reducing IRF1 activity [174].

### 5.4. OAS1

OAS1 (2′-5′-oligoadenylate synthetase 1) is a critical component of the immune response, particularly in antiviral defense, and is regulated by IFN-γ [139]. Distinct alternatively spliced isoforms of OAS1 have been reported in a recent paper highlighting how each isoform has its unique functional properties and enzymatic activities that modulate the immune response. IFN-γ signaling induces OAS1 expression, and the splicing of OAS1 influences how effectively it can mediate antiviral activities. Some isoforms have enhanced enzymatic activity, allowing for a stronger immune response, while others may be less effective. In another article, genetic variations in OAS1 splicing are shown to influence the severity of the disease, highlighting how the differential regulation of OAS1 by IFN-γ can lead to varied immune outcomes. Together, these studies suggest that the regulation of OAS1 splicing by IFN-γ plays a pivotal role in controlling the strength of the antiviral response, influencing susceptibility to viral infections and potentially impacting broader immune regulation in conditions like cancer [175].

There is mounting evidence suggesting that *OAS1* may be an effector gene influencing COVID-19 severity [176]. A recent study shows how two typical human OAS1 gene polymorphisms (rs1131454 and rs10774671) are abundant in European and African COVID-19 patients who need to be hospitalized [177]. These variations increase the expression of an OAS1 transcript that has an early stop codon and undergoes NMD, thus reducing OAS1 protein synthesis which impairs the elimination of viruses. Similarly, Frankiw et al. found that the deletion of a commonly employed splice site in the mouse *Oas1g* gene results in higher Oas1g expression and an improved antiviral action, whereas splicing at the site produces an NMD-targeted transcript [178].

### 5.5. CD20

A splice variant of the B cell lineage membrane receptor CD20 known as D393-CD20 differs from the wild-type CD20 transcript in that it is missing 501 nucleotides in exons 3 to 7 and thus codons 37 to 204 [179,180]. IFN-γ secreting D393-CD20 translation results in a cytosolic protein that lacks both the extracellular region and the majority of the four transmembrane-spanning domains. Typical resting B cells lack D393-CD20 mRNA, whereas different malignant or altered B cells do not. Furthermore, cancerous B cells from relapsed patients who had previously received treatment with the anti-CD20 therapeutic monoclonal antibody Rituximab were shown to have high expression of the D393-CD20 protein. Consequently, it was shown that this spliced mRNA has the capability to be a tumor-related antigen due to its selective expression in leukemic B cells and the production of IFN-γ generating D393-CD20 in resistant lymphomas [179,180].

### 5.6. TrpRS

Aminoacyl-tRNA synthetases represent a crucial category of enzymes responsible for the covalent attachment of amino acids to their corresponding tRNAs [181]. A remarkable illustration of functional diversification is found in human tyrosyl-tRNA synthetase (TyrRS), which undergoes processing by the extracellular protease leukocyte elastase, leading to the release of two distinct fragments: one exhibiting IL-8-like cytokine and pro-angiogenic properties (termed mini-TyrRS) and the other demonstrating EMAP II-like cytokine functionality (designated as the C-domain) [182,183,184]. Notably, the unprocessed, full-length enzyme is devoid of these cytokine activities [185,186]. The closely homologous human enzyme tryptophanyl-tRNA synthetase (TrpRS) has acquired anti-angiogenic properties in the naturally occurring, truncated splice variant known as mini-TrpRS, whereas the full-length, native protein does not exhibit such properties [187,188,189]. This truncated variant, in conjunction with the full-length enzyme, exhibited significant up-regulation in response to IFN-γ [187,190]. Among tRNA synthetases characterized by expanded functionalities, human TrpRS was among the initial subjects of investigation. Preliminary research established that TrpRS undergoes limited proteolytic cleavage and is secreted from the bovine pancreas [191,192,193].

## 6. Therapeutic Perspectives

Current gaps in our knowledge about alternative splicing and IFN-γ activity (limited understanding of how alternative splicing modulates the IFN-γ pathway; incomplete characterization of splicing events in immune cells; minimal investigation of neoepitopes from aberrant RNA splicing; IFN-γ’s therapeutic potential not fully explored) highlight the need for deeper research on how splicing variations impact IFN-γ signaling and immune responses. The role of alternative splicing in modulating immune response is an emerging area of inquiry that holds great potential to yield significant insights [194]. While the role of alternative splicing has been extensively characterized in malignant cells [195,196,197], its biological role in immune cell populations and its therapeutic potential in the context of IFN-γ immunotherapy remains relatively underexplored [198]. Despite its critical role in cancer immunotherapy, the full range of applications of IFN-γ-based immunotherapy strategies as anticancer medications has yet to be fully exploited [199]. A comprehensive understanding of the alternative splicing events that may occur within immune cells located in the tumor microenvironment could enhance and broaden the toolkit of immunotherapeutic strategies in oncological treatment. For instance, the employment of neoepitopes derived from mRNA splicing as prospective targets for anti-cancer immunotherapeutic interventions and/or vaccination may enable a broader cohort of patients to benefit from these approaches [2,84,199].

Promising splicing-based molecular approaches have been discussed elsewhere [200,201] and they include:(a)Compounds that influence alternative splicing, including Spliceostatin, Sudemycins, and FD-895, which exert direct effects on the spliceosome [202].(b)Isoginkgetin, a splicing inhibitor that functions by obstructing the recruitment of U4/U5/U6 small nuclear ribonucleoproteins (snRNPs) [203].(c)Small molecules that modulate splicing factors by targeting their regulatory kinases [204].(d)Sulfonamides like Indisulam (E7070) can diminish the expression of splice factor RBM39 through targeted proteasomal degradation [142,205,206,207,208].

Given the potential of IFN-γ in cancer immunotherapy, developing RNA-based therapies to tweak IFN-γ activity might prove to be a useful strategy. This is particularly relevant considering previous research showing IFN-γ’s critical role in treatments for various conditions, including its potential to improve progression-free survival in ovarian cancer when combined with cyclophosphamide and cisplatin [194].

## 7. Conclusions

Further investigations aimed at delineating altered splicing events and splicing factors associated with cancer are likely to be of paramount importance for developing future targeted therapeutic interventions. The identification of mis-spliced aberrant proteins generated during anti-cancer therapies could yield critical insights into the mechanisms underlying treatment resistance [199]. Therefore, enhanced comprehension of alternative splicing in oncological contexts is expected to unveil a diverse array of novel therapeutic strategies, potentially transforming our approach to cancer treatment by leveraging the intricate mechanisms of immune response and cellular molecular processes [2]. Given that some of these strategies involve IFN-γ, the development of RNA-based drugs to tweak IFN-γ activity in cancer might provide novel avenues for therapeutic intervention.

## Figures and Tables

**Figure 1 cancers-17-00594-f001:**
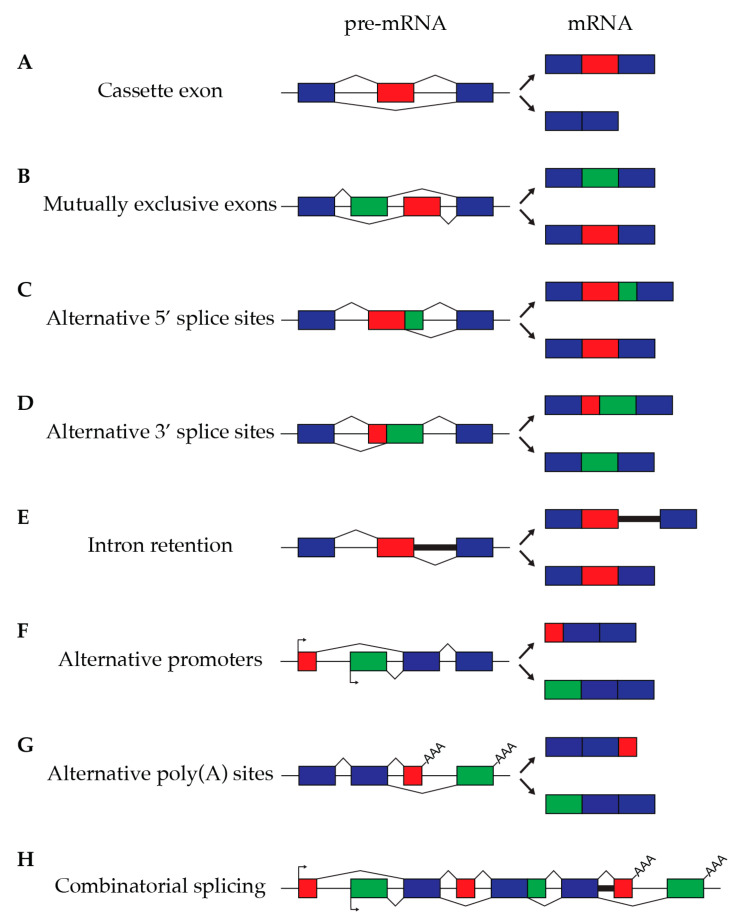
Different types of alternative splicing. Exons are represented by boxes, and introns are represented by lines. Constitutive exons are shown in blue, while alternative exons are shown in red and green. Promoters are indicated with arrows, while different polyadenylation sites are indicated with AAA. There are seven major alternative splicing modes that represent binary choices: cassette exons or exon skipping (**A**), mutually exclusive exons (**B**), alternative 5′ splice sites (**C**), alternative 3′ splice sites (**D**), intron retention (**E**), alternative promoters (**F**), and alternative polyadenylation sites (**G**). Multiple alternative splicing events can also appear in different combinations on the same pre-mRNA (**H**), giving rise to a large repertoire of mRNAs originating from a single pre-mRNA.

**Figure 2 cancers-17-00594-f002:**
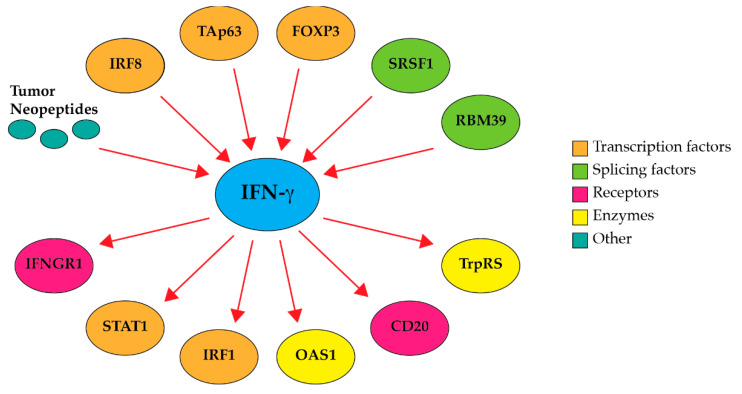
Alternative splicing modulates the activity of the IFN-γ pathway. Schematic representation of upstream regulators and downstream effectors of IFN-γ activity discussed in this review.

**Table 1 cancers-17-00594-t001:** Alternative splicing modulates the activity of upstream regulators of IFN-γ.

Factor	Function	Splicing Event (s)
AMZ2P1-neo; MZT2B-neo	MHC-bound tumor neopeptides	Exon skipping; alternative splice sites; alternative first exons; intron retention
IRF8	Transcription factor	Exon skipping; alternative first exon
Tap63	Transcription factor	Exon skipping
FOXP3	Transcription factor	Exon skipping
SRSF1	Splicing factor	SRSF1-regulated RhoH expression
RBM39	Splicing factor	RBM39-regulated IRF3 expression; inclusion of poison exon in RBM39

**Table 2 cancers-17-00594-t002:** Alternative splicing modulates the activity of downstream effectors of the IFN-γ pathway.

Factor	Function	Splicing Event (s)
IFNGR1	Cell surface receptor subunit	Potential exon skipping
STAT1	Transcription factor	Alternative C-terminal domains
IRF1	Transcription factor	Exon skipping
OAS1	Antiviral defense	Poison exon
CD20	Cell surface receptor	Exon skipping
TrpRS	Tryptophanyl-tRNA synthetase	Truncated splice variants

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
