# Peer review of "Alternative Splicing as a Modulator of the Interferon-Gamma Pathway"

_cancers, 2025, doi:10.3390/cancers17040594_

Round 1
Reviewer 1 Report
Comments and Suggestions for Authors
The review by P. Suri et al. summarizes the processes of alternative splicing that affect the functions of IFN-gamma or are initiated by this protein.
The text is quite good and the logic of the paper is clear.
Major issues
1. The main shortcoming of the paper is the lack of graphical/illustrative material. The authors must visualise the described upstream and downstream INF processes and how they are modified by splicing.
2. FoxP3 can also be spliced to delta7 and delta2delta7 variants, which have an effect on regulatory T cells. This information is missing from the papers.
Minor issues
3. Please introduce the Introduction section.
4. References throughout the text must be merged.
My recommendation is for a major revision of the work mainly in terms of illustrations.
Author Response
1. The main shortcoming of the paper is the lack of graphical/illustrative material. The authors must visualise the described upstream and downstream INF processes and how they are modified by splicing.
Thank you for pointing this out. We have now created a new figure (Figure 2) that addresses this issue and complements Table 1 and Table 2.
2. FoxP3 can also be spliced to delta7 and delta2delta7 variants, which have an effect on regulatory T cells. This information is missing from the papers.
We have now expanded the FoxP3 section (section 4.4), which now includes a discussion of the Δ7 and Δ2Δ7 splice variants.
3. Please introduce the Introduction section.
The manuscript now has an Introduction section (section 1).
4. References throughout the text must be merged.
References are now merged.
Reviewer 2 Report
Comments and Suggestions for Authors
This review explores how alternative splicing modulates interferon-gamma (IFN-γ), a key immune protein involved in fighting cancer and infections by activating immune cells. The process of alternative splicing allows genes to produce different protein variants, which can either enhance or suppress IFN-γ’s effects on immune responses by affecting signaling pathways. While the role of IFN-γ is well-established, the impact of its spliced variants is not fully understood. The review examines how this modulation could provide insights for developing new treatments for cancer, autoimmune diseases, and infections.
Concerns:
1. While the abstract introduces the idea of alternative splicing modulating IFN-γ function, it lacks specific examples or details on how alternative splicing affects signaling proteins or pathways. Providing a brief mention of key examples could add depth.
2. It may help to mention current gaps in knowledge and how the review aims to address these. This would create a clearer roadmap for the reader about the manuscript's contribution to the field.
3. The review alludes to the modulation of IFN-γ's function via alternative splicing but does not specify the mechanisms. A discussion of specific mechanisms, such as changes in receptor affinity, signaling pathway divergence, or protein isoforms, would enhance the clarity of the manuscript's focus.
4. The last section of "Therapeutic Perspectives" can be divided into several subsections based on diseases.
Author Response
1. While the abstract introduces the idea of alternative splicing modulating IFN-γ function, it lacks specific examples or details on how alternative splicing affects signaling proteins or pathways. Providing a brief mention of key examples could add depth.
Response: Unfortunately, there is a 250-word limit for the abstract. The current abstract is already 244 words. If we were to add specific examples to the abstract, we would have to eliminate other parts. We feel that the current abstract is a good compromise, as it is a general overview of the manuscript.
2. It may help to mention current gaps in knowledge and how the review aims to address these. This would create a clearer roadmap for the reader about the manuscript's contribution to the field.
Response: We have expanded section 6 which now also mentions current gaps in knowledge.
3. The review alludes to the modulation of IFN-γ's function via alternative splicing but does not specify the mechanisms. A discussion of specific mechanisms, such as changes in receptor affinity, signaling pathway divergence, or protein isoforms, would enhance the clarity of the manuscript's focus.
Response: We have expanded the relevant sections to include discussion on the mechanism, when possible. Please note that, as far as the splicing mechanism by which specific splice variants are generated, very little is known. Hence gaps in our knowledge and the need for this review.
4. The last section of "Therapeutic Perspectives" can be divided into several subsections based on diseases.
Response: We expanded the last section, which is also now split in 2 with a separate Conclusions section (section 7). However, as the focus of the review is on alternative splicing rather than specific diseases, we feel that the section on Therapeutic Perspectives should focus on splicing rather than specific diseases.
Round 2
Reviewer 1 Report
Comments and Suggestions for Authors
The authors have addressed all my initial concerns. I recommend to accept the manuscript.
Reviewer 2 Report
Comments and Suggestions for Authors
The authors fairly addressed my previous concerns.